DATA RELEASE

# Urban malaria vector dynamics in Accra, Ghana

Abdul Rahim Mohammed Sabtiu[1], Yaw Akuamoah-Boateng[1],
Christopher Mfum Owusu-Asenso[1], Anisa Abdulai[1], Isaac Kwame Sraku[1],
Daniel Kodjo Halou[2], Richard Tettey Doe[1], Emmanuel Nana Boadu[1],
Sebastian Kow Egyin Mensah[1], Judith Dzifa Azumah[3],
Sarkodie Adu-Barima[1], Lourees Esi Awotwe[1], Alberta Walker[1],
Stephina Adjoa Yanney[1], Abena Ahema Ebuako[1],
Nana Aba Sertorwu Eyeson[1], Akua Aboagyewaa Appiah[1],
Bright Churchill Obeng[1], Godfred Amoateng[1], Grace Arhin Danquah[1],
Nutifafa Efui Abusah[1], Edwin Edem Agbotah[1], Ruth Owusu Kwarteng[1],
Jemima Boateng[1], Isaac Amankona Hinne[4], Cornelia Appiah-Kwarteng[1],
Akua Oben Forson[1], Fred Aboagye-Antwi[1] and Yaw Asare Afrane[1,*]

1 Centre for Vector-Borne Disease Research, Department of Medical Microbiology, University of Ghana Medical School, Accra, Ghana
2 Department of Vector Biology, Liverpool School of Tropical Medicine, UK
3 United Graduate School of Drug Discovery and Medical Information Science, Gifu University, Gifu, Japan
4 Department of Biochemistry and Molecular Biology, College of Agriculture, Biotechnology and Natural Resources, University of Nevada, Reno, Nevada, USA

## ABSTRACT

Urban malaria is an emerging challenge in sub-Saharan Africa, driven by unplanned urbanization, irrigation, and vector adaptation; yet data on urban vectors, their diversity and malaria transmission potential are limited. We assessed *Anopheles gambi*ae s.l. abundance, species composition, and behavior in Accra, Ghana, during dry and rainy seasons of 2022 to 2024 across fifteen sites representing different socioeconomic settings. A total of 20,945 host-seeking and 1,613 resting *Anopheles* mosquitoes were collected. Abundance was highest in irrigation and peri-urban sites, and lowest in low socioeconomic areas. *An. gambiae s.s.* dominated host-seeking populations, while *An. coluzzii* dominated resting ones. Findings highlight irrigation and peri-urban areas as hotspots, requiring targeted surveillance and control.

**Subjects** Ecology, Biodiversity, Taxonomy

**Submitted:** 24 September 2025

* Corresponding author. Email: yafrane@ug.edu.gh

Preprint submitted at https://doi.org/10.60763/africarxiv/10163

Included in the series: *Vectors of human disease* (https://doi.org/10.46471/GIGABYTE_SERIES_0002)

## BACKGROUND AND CONTEXT

Urban areas have historically experienced lower malaria prevalence compared to rural settings, primarily attributed to reduced malaria vector populations resulting from fewer preferred breeding habitats and altered environmental conditions [1, 2]. However, recent epidemiological trends indicate a concerning shift, with a growing burden of malaria transmission in Sub-Saharan African cities. Data from the WHO World Malaria Report 2023 indicate an approximate 12% increase in urban malaria cases across fifteen African countries, highlighting the growing influence of rapid urbanization, unplanned environmental modification, and vector adaptation to polluted and atypical breeding habitats on malaria transmission within urban settings [3, 4].

Rising urban malaria in Sub-Saharan Africa is driven by rapid, unplanned urbanization and irrigated urban farming [5–7]. Poor planning has created new *Anopheles* breeding sites including poorly drained areas, construction zones, informal settlements, and water-holding containers [8, 9]. *Anopheles* species have also adapted to urban environments, with populations now thriving in polluted habitats once deemed unsuitable [9–14]. Urban malaria poses a major public health concern as Sub-Saharan Africa undergoes rapid urbanization. By 2050, about 60% of the African population will live in cities, where poor planning and weak health systems foster sustained transmission in dense populations [15].

Despite the threat of urban malaria to public health, data on urban malaria vector dynamics, species diversity, and behavioral patterns remain limited. Existing surveillance programs, largely designed for rural settings, are insufficient for capturing the complexities of urban transmission shaped by environmental factors, human movement, and activities. This study seeks to address these critical knowledge gaps by employing standardized entomological surveillance methods to characterize *Anopheles* populations in urban environments, with the ultimate goal of informing evidence-based malaria vector control strategies tailored to the unique challenges of urban malaria.

## METHODS

### Study sites

This study was conducted at fifteen sites in Accra, Ghana, spanning six socio-economic and environmental categories. Twelve sites were in the urban zone, classified as irrigated urban farming (IUF), low (LS), middle (MS), and high (HS) socio-economic areas, port of entry (PE) and industrial site (IS). Three peri-urban (PU) sites capturing the transitional dynamics between urban and rural settings were also included.

Low socio-economic (LS) areas were characterized by high housing density, poorly structured housing, predominantly unpaved roads, open drains that create breeding sites for mosquitoes, and poor waste management with frequent refuse accumulation. Middle socio-economic (MS) areas generally had moderate housing density, mixed housing structures, a combination of paved and unpaved roads, and both open and closed drainage systems. High socio-economic (HS) areas were characterized by well-structured houses, lower housing density, predominantly paved roads, and organized drainage and waste management systems. Irrigated urban farming (IUF) sites consisted of irrigated vegetable farming areas with irrigation channels and standing water that provide potential mosquito breeding habitats. The port of entry (PE) was the main international airport area, and surrounding areas including residential houses. This site was included because invasive mosquito species may be introduced through activities associated with the nearby airport. Industrial sites (IS) was an area within the city of Accra that had industries where their effluents could create breeding habitats for mosquitoes. Peri-urban (PU) sites were characterized by lower housing density, with new housing developments. This design enabled comparative analysis of vector densities and species distribution across different environment and socio-economic contexts (Figure 1).

The city of Accra, Ghana's capital, lies within the coastal savanna zone. The 2021 census reported 2.56 million inhabitants, a 24% increase since 2010, across 225.67 km² [15]. The city experiences a bimodal rainfall pattern averaging 730 mm annually with the major rainfall falling between April to June and the minor rainy season in September to October. Mean temperatures of 27.6 °C and relative humidity ranging from 65% in the afternoon to

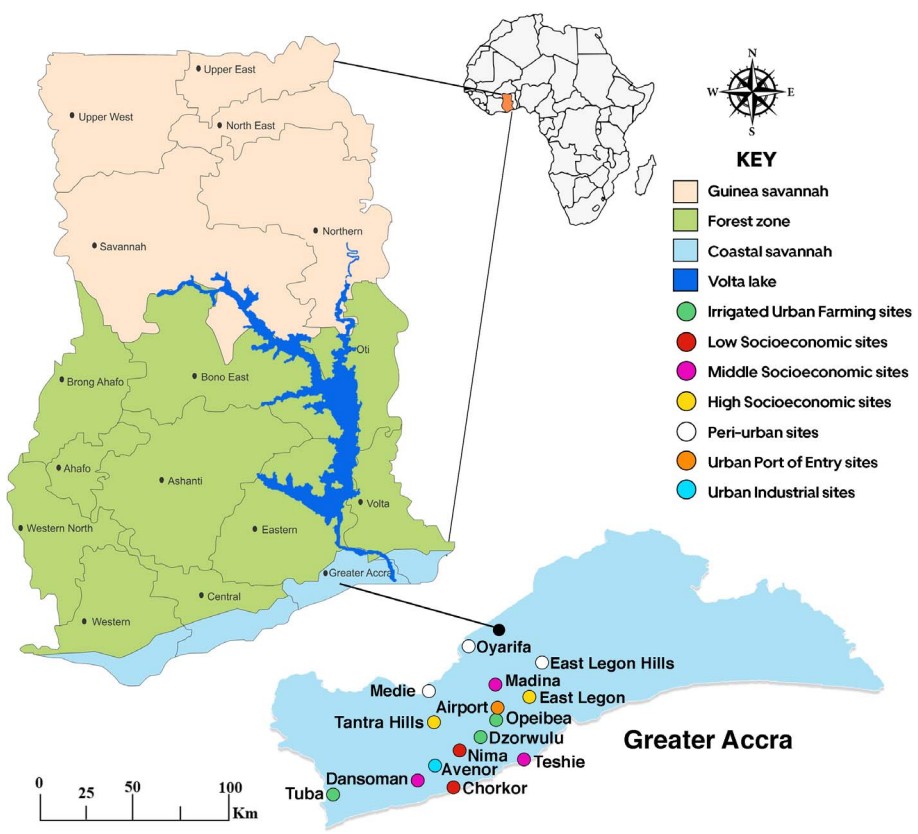

**Figure 1.** Map of Ghana showing study sites.

95% at night. Poor drainage combined with rainfall creates stagnant water bodies, and together with warm, humid conditions, sustains mosquito breeding and year-round malaria transmission across the study sites.

## Collection of host-seeking and resting mosquitoes

Mosquito collections were conducted during the dry (Jan–Mar) and rainy (April–Aug) seasons of 2022–2024 using human landing catches (HLC) [16] and Prokopack (PPK) aspirators. Ten study sites, two from each category (IUF, LS, MS, HS, PU), were selected for HLC. In each site, 12 houses were selected for HLC, with four sampled outdoors on three consecutive nights. Two trained volunteers collected host-seeking mosquitoes from 18:00–06:00 using falcon tubes and flashlights in each house. Specimens were collected hourly, killed (–20 °C freezer or chloroform), stored in labeled silica gel containing Eppendorf tubes, and biting times classified as early evening (18:00–21:00), late evening (00:00–04:00), or early morning (05:00–06:00) [17].

Before HLC mosquito collection, all volunteers were screened for malaria and provided with prophylactic medication consisting of doxycycline hyclate (Vibramycin®; Pfizer Inc., New York, NY, USA) to minimize the risk of infection. They were instructed to report any malaria-like symptoms promptly for immediate testing and treatment. Follow-up malaria testing was performed four weeks after the collection period. Furthermore, volunteers

**Table 1.** Spatio-temporal distribution of host-seeking *An. gambiae* s.l. collected from Accra.

| Study site categories | Study sites | Dry | Rainy | Total |
|---|---|---|---|---|
| IUF | Tuba | 2409 | 6061 | 8470 |
| | Opeibea | 658 | 1096 | 1754 |
| LS | Nima | 71 | 258 | 329 |
| | Chorkor | 56 | 210 | 266 |
| MS | Dansoman | 80 | 1002 | 1082 |
| | Teshie | 389 | 1400 | 1789 |
| HS | Tantra Hill | 403 | 1326 | 1729 |
| | East Legon | 195 | 1579 | 1774 |
| PU | Medie | 223 | 1255 | 1478 |
| | Oyarifa | 582 | 1678 | 2260 |
| Total | | 5,066 | 15,865 | 20,931 |

were recruited from the study communities to assist with field activities and provide safety supervision throughout the night.

Resting mosquitoes were collected with PPK aspirators from 15 study sites. Each morning from 05:30 am–08:00 am, five houses were randomly selected and sampled for resting mosquitoes for three days. Households were instructed to keep doors and windows closed overnight. Mosquitoes were aspirated from ceilings, walls, and furniture, then transferred to holding cups and transported for laboratory identification [18].

## Data validation and quality control

All mosquitoes were sorted by genus (*Anopheles*, *Culex*, *Aedes*), sex, and gonotrophic stage (unfed, fed, semi-gravid, gravid). *An. gambiae* s.l. (ncbitaxon:7165) were identified morphologically under a stereomicroscope using Coetzee's key [19]. A sub-sample of the host seeking mosquitoes (4,524) and resting mosquitoes (890) were discriminated using rDNA-PCR [20]. *Anopheles gambiae* s.s. and *An. coluzzii* were further identified using PCR-RFLP [21].

## RESULTS

## Abundance and distribution of host-seeking *An. gambiae* s.l. collected in Accra

A total of 20,945 *Anopheles* mosquitoes consisting of *An. gambiae* s.l. (99.93%, 20,931/20,945), *An. funestus* (0.02%, 4/20,945), *An. rufipes* (0.02%, 4/20,945) and *An. pharoensis* (0.03%, 6/20,945) were collected using HLC. Of the 20,931 *An. gambiae* s.l., 75.81% (15,865/20,931) were obtained during the rainy season, while 24.20% (5,066/20,931) were collected in the dry season.

By site category, the greatest abundance of *An. gambiae* s.l. was recorded in IUF areas, which accounted for 48.85% (10,224/20,931) of the total catch. This was followed by PU sites, contributing 17.86% (3,738/20,931). HS site category recorded 16.73% (3,503/20,931), MS sites accounted for 13.72% (2,871/20,931), whereas LS sites contributed the lowest proportion 2.84% (595/20,931), of the overall *An. gambiae* s.l. population collected.

At the individual site level, the highest numbers were observed in Tuba (40.47%, 8,470/20,931) within the IUF category, followed by Oyarifa (10.80%, 2,260/20,931) in the PU category. The lowest abundance was recorded in Chorkor (1.27%, 266/20,931) within the LS category (Table 1).

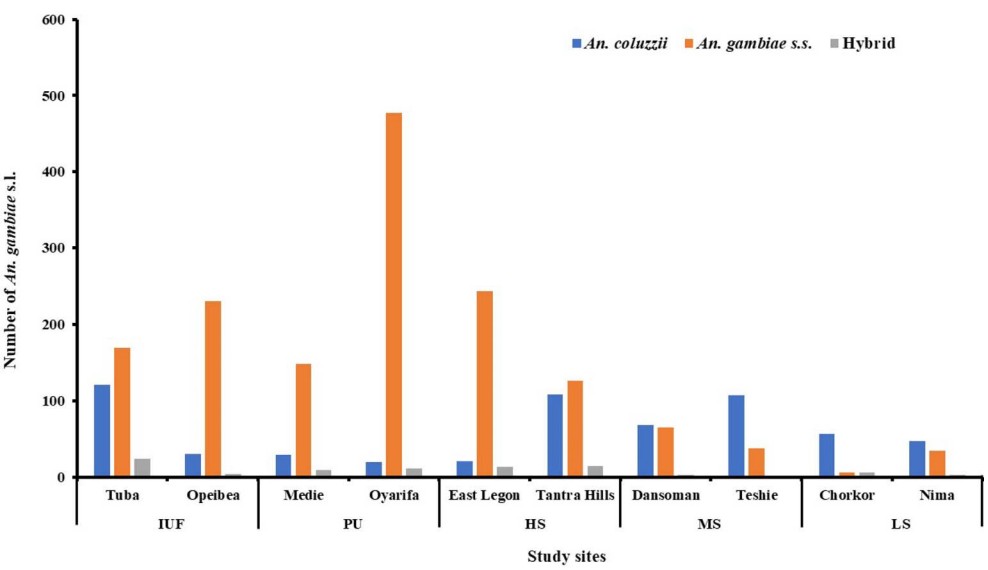

**Figure 2.** Abundance and distribution of *An. gambiae* s.l. sibling species across sites.

## Species discrimination of Host-seeking *An. gambiae* s.l. collected in Accra

Of the 4,524 *An. gambiae* s.l. sub-sampled across the study sites, 59.70% (2,701/4,524) were identified as *An. gambiae* s.s., 33.80% (1,529/4,524) *An. coluzzii* and 6.50% (294/4,524) were hybrids of the two species.

At the site category level, *An. gambiae* s.s was the most dominant species identified in IUF (62.44%, 1,009/1,616), PU (73.03%, 861/1,179), MS (48.31%, 329/681) and HS sites (55.72%, 443/795). However, in LS sites *An. coluzzii* was the most dominant species identified (55.72%, 173/253) (Figure 2).

## Abundance and seasonal distribution of resting *An. gambiae* s.l. across study sites in Accra

A total of 1,613 outdoor and indoor resting *An. gambiae* s.l. were collected across the study sites in Accra. Of these, 54.74% (883/1,613) were collected during the rainy season and 45.26% (730/1,613) during the dry season. Outdoor collections (66.34%, 1,070/1,613) were consistently higher than indoor collections (33.66%, 543/1,613). The IUF sites recorded the highest abundance, contributing 68.57% (1,106/1,613) of all mosquitoes collected. The LS sites had 9.05% (146/1,613), MS sites 8.49% (137/1,613), and PU sites 9.55% (154/1,613). By contrast, PE (2.48%, 40/1,613), HS (1.49%, 24/1,619) and IS (0.37%, 6/1,613) recorded the lowest collections. Overall, mosquito abundance peaked during the rainy season, particularly in Tuba, where both indoor and outdoor resting catches were comparatively high (Table 2).

**Table 2.** Seasonal abundance and resting behaviour of *An. gambiae* s.l. across study sites.

| Site categories | Study sites | Dry Indoor | Dry Outdoor | Rainy Indoor | Rainy Outdoor | Total |
|---|---|---|---|---|---|---|
| IUF | Tuba | 26 | 523 | 224 | 228 | 1001 |
| | Opeibea | 17 | 15 | 42 | 29 | 103 |
| | Dzorwulu | 0 | 2 | 0 | 0 | 2 |
| LS | Nima | 7 | 26 | 26 | 56 | 115 |
| | Chorkor | 5 | 4 | 13 | 9 | 31 |
| MS | Dansoman | 6 | 29 | 28 | 11 | 74 |
| | Teshie | 7 | 7 | 22 | 16 | 52 |
| | Madina | 0 | 1 | 5 | 5 | 11 |
| HS | Tantra Hill | 4 | 0 | 10 | 6 | 20 |
| | East Legon | 0 | 4 | 0 | 0 | 4 |
| PU | Medie | 7 | 6 | 35 | 27 | 75 |
| | Oyarifa | 11 | 8 | 32 | 24 | 75 |
| | East Legon Hills | 0 | 4 | 0 | 0 | 4 |
| PE | Airport | 0 | 5 | 16 | 19 | 40 |
| IS | Avenor | 0 | 6 | 0 | 0 | 6 |
| Total | | 90 | 640 | 453 | 430 | 1613 |

**Figure 3.** Species discrimination and distribution of resting *An. gambiae* s.l. across study sites.

## Species discrimination and distribution of resting *An. gambiae* s.l. across study sites

*An. coluzzii* was the predominant species across both seasons [rainy (52.32%, 462/883), dry (55.21%, 403/730)] and resting sites [indoor (33.87%, 293/865), outdoor (66.13%, 572/865)]. Its abundance was higher outdoors than indoors and peaked during the rainy season. *An. gambiae s.s.* was less abundant (37.45%, 604/1,613), while hybrids were the least identified species in both seasons (Figure 3).



## RE-USE POTENTIAL

This dataset provides detailed information on *An. gambiae* s.l. populations across urban and peri-urban sites in Accra, Ghana. It captures seasonal variation, species composition, and indoor–outdoor resting and host-seeking behavior, offering insights into urban malaria transmission dynamics.

Researchers can use the data to assess spatial and seasonal patterns, evaluate the influence of socio-environmental factors, and improve surveillance methods in urban settings. Public health programs can apply it to design targeted, evidence-based vector control strategies, while also serving as a baseline for studying mosquito adaptation and informing broader vector ecology and disease surveillance in urban Africa.

## DATA AVAILABILITY

The dataset presented in this study is accessible through the GBIF repository [22].

## EDITOR'S NOTE

This paper is part of a series of Data Release articles working with GBIF and supported by TDR, the Special Program for Research and Training in Tropical Diseases, hosted at the World Health Organization [23].

## ABBREVIATIONS

GBIF – Global Biodiversity Information Facility; HLC – Human Landing Catches; HS – High Socioeconomic; IS – Industrial site; IUF – Irrigated Urban Farming; LS – Lower Socioeconomic; MS – Middle Socioeconomic; PE – Port of entry; PPK – Prokopack aspirator; PU – Peri-urban.

## DECLARATIONS

### Ethical approval

This study received scientific and ethical approval from the Ethical and Protocol Review Committee (EPRC), College of Health Sciences, University of Ghana, Korle-Bu Campus. Verbal and written informed consent was obtained from community opinion leaders and household heads at all selected study sites prior to mosquito sampling.

### Consent for publication

Not applicable.

## COMPETING INTEREST

The authors declare no competing interests.

## Authors' contributions

YAA conceived, designed, and supervised the study. ARMS, YAB, CMO-A, AA, IAK, JDA, SAB, LEA, AW, SPY, NASE, AAA, RD, ENB, SKEM, BCO, GA, GAD, NEA, ROK, JB, DKH, IAH collected the data and contributed to its analysis. ARMS and YAB drafted the manuscript. All authors reviewed and approved the final version.



## Funding

This study was supported by grants from the National Institutes of Health (R01 A1123074, RO3 AI186018, and D43 TW 011513). The funding agencies had no role in the design of the study, data collection, analysis, interpretation, or manuscript preparation.

## Acknowledgements

We are grateful to the participating communities for their cooperation and for allowing mosquito sampling within their homes and surroundings. We also extend our appreciation to the Centre for Vector-borne Disease Research (CVBDR) team, Department of Medical Microbiology, University of Ghana Medical School, for their invaluable support throughout the project.

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
