## [Editor Report]

Editor’s AssessmentIn Africa urban malaria is an increasing challenge, there is however very limited data on Malaria vector species urban ecology, hindering modelling research and control strategies. This paper is one of a series of Data Release papers in GigaByte supported by TDR and the WHO describing datasets hosted in GBIF to tackle these data gaps in vectors of human disease data. This paper presents a dataset on Anopheles gambiae s.l. abundance, species composition, and behaviour in Accra, Ghana, during dry and rainy seasons of 2023 and 2024. Sampling conducted at fifteen sites in Accra, Ghana, spanning six socio-economic and environmental categories, with a total of 21,064 specimens collected across all study sites during the sampling period. Peer review and data auditing found the data to be well validated. The information contained can serve as a resource for studies focused on assessing transmission risks, vector control strategies, disease surveillance and a broader comprehension of Culex and Aedes mosquito ecology in the various ecological zones of Ghana.Editor’s AssessmentIn Africa urban malaria is an increasing challenge, there is however very limited data on Malaria vector species urban ecology, hindering modelling research and control strategies. This paper is one of a series of Data Release papers in GigaByte supported by TDR and the WHO describing datasets hosted in GBIF to tackle these data gaps in vectors of human disease data. This paper presents a dataset on Anopheles gambiae s.l. abundance, species composition, and behaviour in Accra, Ghana, during dry and rainy seasons of 2023 and 2024. Sampling conducted at fifteen sites in Accra, Ghana, spanning six socio-economic and environmental categories, with a total of 21,064 specimens collected across all study sites during the sampling period. Peer review and data auditing found the data to be well validated. The information contained can serve as a resource for studies focused on assessing transmission risks, vector control strategies, disease surveillance and a broader comprehension of Culex and Aedes mosquito ecology in the various ecological zones of Ghana.

---

## [Reviewer Report]

Indicate in the comments box below whether you are happy with the changes made or if the manuscript is unacceptable.Comments on revised manuscriptDear Editor. While the authors of this manuscript have made a significant corrections, addressing most of my observations, there are two not addressed. 1. “Site selection considered development level, drainage, sanitation, and commercial activities” These are very vague descriptions, lacking operational clarity, making it scientifically weak, and which has to be addressed quantitatively. For example, developmental level is a subjective phrase. I cannot tell how development was judged or how different one site was from another site. Without quantitative or categorical criteria it is unverifiable statement. There were no referenced indicators. Without measurable proxies there is no linkage to show how these factors affect vector ecology. The fact that authors did not take these factors into cognizance [e.g., for developmental level (access to electricity, presence of paved roads, household density, socio-economic indices taken directly or from national census data, for drainage (presence of open vs closed rains, etc), sanitation (access to latrines, waste disposal presence/absence of refuse heaps), commercial activity (density of shops, markets, type, etc) make it impossible to accept the rationale for this study within the context of the urban vs peri-urban sites. As a major revision authors should take some of these major factors into account and use them in regression analysis for analysis of their vector abundance and composition. The above concern needs to be addressed by providing relevant information expanding in the methods section or a supplementary table detailing the criteria used for categorization and relevant references. 2. Volunteers for HLC were given prophylaxis. Which brand of drug exactly? That should be reflected, with company, city and country of manufacturing within the methods section.

---

## [Reviewer Report]

Upload additional filesDRR-202509-07-R02/stage_files/DRR-202509-07/Review MS/GBIF-Data-Review-DRR-202509-07.pdfReviewer name and names of any other individual's who aided in reviewer Yannan FanDo you understand and agree to our policy of having open and named reviews, and having your review included with the published papers. (If no, please inform the editor that you cannot review this manuscript.)YesIs the language of sufficient quality?YesPlease add additional comments on language quality to clarify if needed
Are all data available and do they match the descriptions in the paper? NoAdditional CommentsThe Occurence in the GBIF dataset is 5414, but the MS described as 21064 in table 1. Please provide more detail description in the MS.Are the data and metadata consistent with relevant minimum information or reporting standards? See GigaDB checklists for examples <a href="http://gigadb.org/site/guide" target="_blank">http://gigadb.org/site/guide</a>YesAdditional CommentsIs the data acquisition clear, complete and methodologically sound?YesAdditional CommentsIs there sufficient detail in the methods and data-processing steps to allow reproduction?YesAdditional CommentsIs there sufficient data validation and statistical analyses of data quality? Not my area of expertiseAdditional CommentsIs the validation suitable for this type of data?YesAdditional CommentsIs there sufficient information for others to reuse this dataset or integrate it with other data?YesAdditional CommentsAny Additional Overall Comments to the AuthorRecommendationMinor Revision

---

## [Reviewer Report]

Upload additional filesDRR-202509-07-R02/stage_files/DRR-202509-07/Review MS/Review_Report_Sabtiu_et_al_GigaByte_2025.docxReviewer name and names of any other individual's who aided in reviewer Sulaiman S. IbrahimDo you understand and agree to our policy of having open and named reviews, and having your review included with the published papers. (If no, please inform the editor that you cannot review this manuscript.)YesIs the language of sufficient quality?YesPlease add additional comments on language quality to clarify if needed
Minor language revision can make the manuscript better.Are all data available and do they match the descriptions in the paper? NoAdditional Comments"Site selection considered development level, drainage, sanitation, and commercial activities” These are very vague descriptions, lacking operational clarity, making it scientifically weak, and which has to be addressed quantitatively.Are the data and metadata consistent with relevant minimum information or reporting standards? See GigaDB checklists for examples <a href="http://gigadb.org/site/guide" target="_blank">http://gigadb.org/site/guide</a>NoAdditional CommentsPlease see my comment above. Key data on site selection, and which are supposed to be used for correlation with composition and abundance of the mosquitoes are missing.Is the data acquisition clear, complete and methodologically sound?NoAdditional CommentsPlease see my comment above.Is there sufficient detail in the methods and data-processing steps to allow reproduction?NoAdditional CommentsThere is information on samples collection in the methods, but no dedicated section on data processing, and with the contentious section on characteristics of selected sites it is not possible to reproduce what authors did.Is there sufficient data validation and statistical analyses of data quality? NoAdditional CommentsThere is some descriptive analysis of the generated data, but no core statistics were conducted to show the significant difference in species composition between the distinct sites.Is the validation suitable for this type of data?YesAdditional CommentsValidation is important for this kind of data.Is there sufficient information for others to reuse this dataset or integrate it with other data?YesAdditional CommentsThere is a link in reference 21 for this.Any Additional Overall Comments to the AuthorDear Authors. I have read through your manuscript and made observations to make it better. However, you have not provided a design that allow comparison of vector densities/malaria transmission across socio-economic contexts or the ecological sites. Your title include transmission risk, but you did not do screen your mosquitoes for the source of blood, to show they bite humans. Also, you did not screen them for Plasmodium infection. I don't see any evidence of transmission here.RecommendationMinor Revision